# Patterns of Genetic Diversity among Alphasatellites Infecting *Gossypium* Species

**DOI:** 10.3390/pathogens11070763

**Published:** 2022-07-04

**Authors:** Muhammad Mubin, Arzoo Shabbir, Nazia Nahid, Iram Liaqat, Muhammad Hassan, Nada H. Aljarba, Ahmed Al Qahtani, Claude M. Fauquet, Jian Ye, Muhammad Shah Nawaz-ul-Rehman

**Affiliations:** 1Virology Lab, CABB University of Agriculture, Jail Road, Faisalabad 38000, Pakistan; mmubin@uaf.edu.pk (M.M.); arzooshabbir96@gmail.com (A.S.); hassan_coms@yahoo.com (M.H.); 2Department of Bioinformatics and Biotechnology, GC University Faisalabad, Faisalabad 38000, Pakistan; nazianahid@gmail.com; 3Microbiology Laboratory, Department of Zoology, GC University Lahore, Lahore 54000, Pakistan; iramliaq@hotmail.com; 4Department of Biology, College of Science, Princess Nourah Bint Abdulrahman University, Riyadh 11671, Saudi Arabia; nhaljarba@pnu.edu.sa; 5Department of Infection and Immunity, Research Center, King FaisaI Specialist Hospital and Research Center, Riyadh 11564, Saudi Arabia; aqahtani@kfshrc.edu.sa; 6Department of Microbiology and Immunology, College of Medicine, Alfaisal University, Riyadh 11533, Saudi Arabia; 7CIAT, Apdo. Aereo, Cali 6713, Colombia; cfauquet@gmail.com; 8Laboratory of Vector-Borne Diseases, State Key Laboratory of Plant Genomics, Institute of Microbiology, Chinese Academy of Sciences, Beijing 100101, China; jianye@im.ac.cn

**Keywords:** alphasatellite, epidemic, begomoviruses, genetic recombination, diversity, cotton

## Abstract

Alphasatellites are small single-stranded circular DNA molecules associated with geminiviruses and nanoviruses. In this study, a meta-analysis of known alphasatellites isolated from the genus *Gossypium* (cotton) over the last two decades was performed. The phylogenetic and pairwise sequence identity analysis suggested that cotton-infecting begomoviruses were associated with at least 12 different alphasatellites globally. Three out of twelve alphasatellite were associated with cotton leaf curl geminiviruses but were not isolated from cotton plants. The cotton leaf curl Multan alphasatellite, which was initially isolated from cotton, has now been reported in several plant species, including monocot plants such as sugarcane. Our recombination analysis suggested that four alphasatellites, namely cotton leaf curl Lucknow alphasatellites, cotton leaf curl Multan alphasatellites, Ageratum yellow vein Indian alphasatellites and Ageratum enation alphasatellites, evolved through recombination. Additionally, high genetic variability was detected among the cotton-infecting alphasatellites at the genome level. The nucleotide substitution rate for the replication protein of alphasatellites (alpha-Rep) was estimated to be relatively high (~1.56 × 10^−3^). However, unlike other begomoviruses and satellites, the first codon position of alpha-Rep rapidly changed compared to the second and third codon positions. This study highlights the biodiversity and recombination of alphasatellites associated with the leaf curl diseases of cotton crops.

## 1. Introduction

Alphasatellites, betasatellites and deltasatellites are small single-stranded (ss) circular DNA molecules associated with nanoviruses and geminiviruses [1]. Alphasatellites are capable of replicating their own genome, while betasatellites and deltasatellites are trans-replicated by a helper or cognate virus [2]. Geminialphasatellites (subfamily *Geminialphasatellitinae*) are single-stranded circular molecules of ~1370–1400 nucleotides (nts) that are associated with begomoviruses and mastreviruses from the family *Geminiviridae* [3,4]. In terms of genome organization and nucleotide composition, alphasatellites are closely related to the R component of nanoviruses, another distinct class of single-stranded DNA viruses [5,6]. However, alphasatellites are relatively large (~1370 nt) compared to the R component (~1000 nts) of nanoviruses. Alphasatellites associated with geminiviruses are considered as non-essential molecules, while betasatellites are pathogenicity determinants. Both alphasatellites and betasatellites are associated with monopartite begomoviruses in Asia and Africa [6,7,8]. The known Old World (OW) and New World (NW) alphasatellites are autonomously-replicating molecules with three distinct regions in their genomes: a nanovirus-related (rather than geminivirus-related) origin of replication, an ORF coding for replication-associated protein (alpha-Rep) and an adenosine-rich region [9,10]. However, the alpha-Rep protein of New World alphasatellites and the OW Ageratum yellow vein Singapore alphasatellite has an insertion of 13 amino acids [11]. NW alphasatellites are phylogenetically distinct (classified under the genera *Clecrusatellite*) and associated with bipartite viruses [11,12]. Recently, alphasatellites of African origin (CLCGezA) were also reported in the Americas [13].

Old World geminialphasatellites play a role in the suppression of gene silencing at the transcriptional and post-transcriptional levels [14,15,16]. They have also been reported to attenuate the symptoms caused by helper viruses [17,18,19]. The experiments conducted on NW alphasatellites revealed that they increased disease severity while negatively affecting viral transmission by *Bemicia tabaci* [12].

The family *Alphasatellitidae* is divided into three subfamilies: *Geminialphasatellitinae*, comprising seven genera; *Petromoalphasatellitinae*, comprising five genera; and *Nanoalphasatellitinae*, comprising six genera [20,21]. All cotton-infecting alphasatellites belong to the *Geminialphasatellitinae* subfamily. The first alphasatellite, “*Cotton leaf curl Multan alphasatellite* (CLCMulA)”, was isolated from cotton crop in 1999 [8]. The name of the satellite was derived from the helper virus, the cotton leaf curl Multan virus (CLCMulV). Later, alphasatellites were also isolated from other plant hosts, including ageratum, papaya, parthenium, potato and chilies [6,22,23,24,25]. In Africa, the cotton leaf curl Gezira alphasatellite (CLCGezA) was reported to infect okra crops [7]. Today, there are 85 recognized species of alphasatellites, along with two tentative species (isolated from insects), that infect different Old World and New World crops [20,21]. Cotton-infecting alphasatellites have also been reported in other crops, such as okra, tobacco and papaya. However, there is currently no precise data on the number of alphasatellite that infect economically important crops, such as cotton.

The diversity of alphasatellites has been explored in several crops, including cotton, tomato and chilies [26,27]. Such surveys are important and provide crucial information; however, the studies conducted still cannot provide a detailed picture of alphasatellite diversity. Therefore, in this study, we used metadata for cotton-infecting alphasatellites from GenBank to carry out a comprehensive phylogenetic and recombination analysis of these satellites.

## 2. Materials and Methods

### 2.1. Alphasatellite Datasets

The full-length nucleotide sequences of 420 alphasatellites associated with cotton leaf curl disease were downloaded from GenBank in FASTA format on 30 December 2021 (Appendix A). The acronyms for the sequences were determined according to [21], and the host name and year of identification were obtained from the descriptions of each molecule. 

### 2.2. Nucleotide Identity Percentage

Full-length sequences were imported into the sequence demarcation tool (SDT) software [28] and aligned with MUSCLE. The pairwise nucleotide identity percentages were calculated and compared to determine the species names, as described previously in [21].

### 2.3. Phylogenetic Analysis

For phylogenetic analysis, all sequences were exported to the MEGA DNA analysis software [29] and aligned using CLUSTAL-W; phylogenetic trees were generated using the maximum likelihood method integrated in the MEGA software (Appendix A). A total of 1000 bootstrap replications were performed to assess the strength of the phylogenetic tree branches. The sequences and GenBank accession numbers of the 420 isolates are provided in Appendix A. The representative molecules from each species/cluster were separated and their Rep genes were predicted using the ApE software (https://jorgensen.biology.utah.edu/wayned/ape/) (accessed on 31 December 2021). Both the full-length and Rep molecules were aligned with MUSCLE, and phylogenetic trees were generated using the maximum likelihood method [29].

### 2.4. Recombination Analysis of Alphasatellites

A recombination analysis of alphasatellite molecules was conducted using the RDP5 software [30]. Briefly, alphasatellite sequences from the different species were imported into the RDP5 software. A probability level of 0.05 was used as a cutoff value for significance. Recombination was also confirmed via BLAST analysis and through the phylogenetic trees of the Rep gene and the common regions of the selected alphasatellite sequences. Recombination of alphasatellites was assessed through six different algorithms: SiScan, RDP, MaxChi, GENECONV, Chimera and BootScan. The *p*-values and breakpoints for different recombination events are provided in Appendix A.

### 2.5. BEAST Analysis of Cotton Leaf Curl Multan Alphasatellites

To estimate the nucleotide substitution rate, the alpha-Rep gene was selected from recombination-free molecules. The alpha-Rep gene was predicted using the ApE software (https://jorgensen.biology.utah.edu/wayned/ape/) (accessed on 31 December 2021). Equal-sized Rep genes (number of taxa = 36) were aligned and exported in Nexus format for further analysis with the Bayesian evolutionary sequence tool (BEAST) [31] (Appendix A). The year of identification of the alphasatellite was recorded from GenBank and added to the Beauti module of the BEAST. Each molecule was partitioned into 3 codon positions for mutation estimates. The BEAST program was set to run for 5 × 10^7^ states with a burnin value of 5%. The log file generated by the BEAST software was run in Tracer (https://github.com/beast-dev/tracer/releases/tag/v1.7.1) (accessed on 31 December 2021) to estimate nucleotide substitutions and the mutation of codon positions.

### 2.6. Pairwise Sequence Comparison Analysis

Pairwise sequence comparison (PASC) was used to analyze the nucleotide diversity of alphasatellites at the species and genus level. Briefly, the full-length genomes (*n* = 420) reported from cotton were analyzed for sequence similarities in the SDT program [28]. The alpha-Reps of all the alphasatellites (*n* = 420) were also isolated for each accession, and their identity percentage values were calculated using the SDT program. A matrix of pairwise sequence identity calculated using SDT (MUSCLE module) was used. The identity percentage values and their frequency distribution were transferred to a spreadsheet to plot the PASC, as described previously in [32].

## 3. Results

### 3.1. Twelve Different Alphasatellites Are Associated with Cotton Leaf Curl Geminiviruses around the World

The meta-analysis of 420 full-length alphasatellite sequences reported in GenBank showed that 12 different alphasatellites were associated with cotton or cotton leaf curl geminiviruses (Table 1). Cotton leaf curl Multan alphasatellite (CLCMulA), the first alphasatellite identified from cotton (a dicot) in Pakistan [8], was also reported in sugarcane (a monocot; Table 1).

The meta-analysis revealed that several alphasatellites, including Ageratum enation alphasatellite (AEA), Ageratum yellow vein Indian alphasatellite (AYVINA), Croton yellow vein mosaic alphasatellite (CrYVMA), Ageratum yellow vein Singapore alphasatellite (AYVSGA), Gossypium Mustilinum symptomless alphasatellite (GMusSLA), Gossypium darwinii symptomless alphasatellite (GDarSLA) and cotton leaf curl Lucknow alphasatellite (CLCLucA), were reported in cotton crop from India and Pakistan. Cotton leaf curl Egypt alphasatellite (CLCEA) was the only alphasatellite reported in cotton from the African continent. These nine alphasatellites are considered “bona fide” cotton-infecting alphasatellites, as they were originally isolated from cotton hosts. Interestingly, out of the nine alphasatellites, four—AEA, AYVINA, AYVSGA and CrYVMA—were previously reported in weeds [25,33]. Three other alphasatellites, namely *cotton leaf curl Gezira alphasatellite* (CLCGezA), *cotton leaf curl Botswana alphasatellite* (CLCBTA) and *cotton leaf curl Saudi Arabia alphasatellite*, were found in Africa and the Middle East. None of these isolates were reported in cotton (Table 1); therefore, we considered them as “putative” cotton-infecting alphasatellites rather than bona fide cotton-infecting alphasatellites.

The Geminialphasatellitinae subfamily comprises seven genera. Members from four of these genera (Ageyesisatellite, Clecrusatellite, Colecusatellite and Gosmusatellite) infect cotton in Asia and Africa (Table 1).

Cotton leaf curl Gezira alphasatellite was thought to be the only alphasatellite present in Africa; however, according to a recent classification and the nucleotide identity analysis we performed here, two more alphasatellites—cotton leaf curl Egypt alphasatellite (CLCEA) and cotton leaf curl Saudi Arabia alphasatellite (CLCSAA)—were also found in association with cotton leaf curl geminiviruses in the Middle East (Table 1 and Table 2) [34]. The identified hosts provided in GenBank suggest that, similarly to CLCMulA, which has largely spread in India and Pakistan, CLCGezA is widespread in Africa and the Middle East (Table 1).

### 3.2. Phylogeny of Cotton-Infecting Alphasatellites

The phylogenetic tree based on the full-length and alpha-Rep molecules clearly resulted in four distinct groups corresponding to four genera (*Colecusatellite*, *Gosmusatellite*, *Clecrusatellite* and *Ageyesisatellite*; Figure 1A,B). The first five alphasatellites (CLCMulA, AEA, CLCLucA, AYVINA and GDarSLA) originated from the Indian subcontinent and were clustered together in the phylogenetic tree (Figure 1). The alphasatellites of African origin (CLCEA, CLCGeA and CLCBTA) occupied a unique position in the phylogenetic tree; they clearly belong to the *Colecusatellite* genus. Interestingly, the phylogenetic tree showed that isolates of CLCMulA and AEA identified in India and Pakistan did not form a monophyletic group. However, all other alphasatellites formed a monophyletic group in the phylogenetic tree (Figure 1A). We were able to detect at least nine different groups of cotton-infecting alphasatellites due to their recombination with either AEA or CLCMulA (Figure 1, clusters a to i). The eight different clusters involving recombination with CLCMulA (a, b, c, d, e, f, g and h) were grouped together in the phylogenetic tree, while cluster i with GDarSLA was segregated. A phylogenetic tree of the Rep gene for all these isolates was also constructed (Figure 1B). As expected, with the exception of clusters a, b and c, the positions of five other clusters in the Rep tree were changed. From the phylogenetic tree, it is clear that CLCLucA acquired its alpha-Rep gene from AEA (Figure 1B, cluster-d). However, the full-length genomes of CLCLucA were segregated with their possible progenitors, i.e., CLCMulA and AEA (Figure 1A, cluster d).

The recombinants were also supported by their unique positions in the phylogenetic tree following the outcome of recombination analysis (Figure 1B and Appendix A). Clusters a, b and c included the CLCMulA and AEA isolates, which recombined with each other (Figure 1B and Appendix A). The isolates of cluster d included two isolates of CLCLucA. Both the molecules had an A-rich region similar to CLCMulA, while the remaining nucleotides (spanning the Rep region) had a high sequence identity with AEA (85–89%; Table 2). The molecules from clusters e and f represented the isolates, which showed either intraspecific recombination or no detectable level of recombination. Meanwhile, clusters g, h and i represented the recombination of CLCMulA with AEA, AYVINA and GDarSLA isolates, respectively. The CLCMulA isolates of cluster i occupied a unique position in the phylogenetic tree due to their recombination with GDarSLA.

Interestingly, apart from AEA, AYVINA, CLCMulA and CLCLucA, none of the other alphasatellites showed any detectable recombination.

The members of the *Colecusatellite*, *Gosmusatellite*, *Clecrusatellite* and *Ageyesisatellite* genera infect cotton crop around the world. CrYVMA has been reported in cotton from Pakistan (GenBank; LT615039), while the cotton leaf curl Lucknow alphasatellite (CLCLucA) has been found in India (GenBank; HQ343234) [35]. This shows that both alphasatellites were locally introduced to cotton from weeds or other sources through their whitefly vector.

### 3.3. Recombinant Alphasatellites of the Genus Colecusatellite Have a High Degree of Sequence Similarity

On the basis of the identity percentages of the full-length genome and alpha-Rep sequences, we found that isolates of CLCMulA and AEA, CLCLucA, AYVINA and GDarSLA were evolutionarily linked with each other at the nucleotide sequence level (Table 2). These isolates exhibited a close range of sequence identity percentages in the upper limits, justifying the current species criteria for alphasatellite classification, which was set at 88% by [21] (Table 2). However, the recombination and higher identity percentages of alpha-Rep (>88%) between the isolates of CLCMulA, AEA, CLCLucA, AYVINA and GDarSLA suggest that their taxonomical status can be revisited and the isolates showing higher identities at the gene level may be merged. For example, at the whole genome level, the isolates of AEA and CLCMulA were 76–90% identical to each other, while at the alpha-Rep level, their nucleotide identity range was 80–90%, indicating that their individual species status should be changed. Similar patterns of sequence similarity were also observed for the isolates of CLCLucA, AYVINA and GDarSLA with CLCMulA (Table 2). By analyzing the PASC values of the full-length alphasatellites and the Rep gene together, it became obvious that the range of identity percentage was increased at the alpha-Rep level (Table 2 and Figure 2). These observations led to the conclusion that due to the variable nature of the A-rich region and recombination in the common region, the identity percentage may go below the species level (Table 2).

### 3.4. Cotton Leaf Curl Multan alphasatellite Has a High Nucleotide Substitution Rate

Interestingly, the nucleotide substitution rate of CLCMulA (1.56 × 10^−3^) was closer to that of known geminialphasatellites (Table 3). The nucleotide substitution rate for the three-codon positions was surprisingly similar to GDarSLA (Table 3), with values of 1.981, 0.516, and 0.501. These values suggested that in CLCMulA, instead of the third codon position (the variable wobble position), the first codon position was more variable.

## 4. Discussion

Cotton (*Gossypium hirsutum* L.) is the main fiber-producing crop in Pakistan, and cotton leaf curl disease (CLCuD) is a major reason for low yield. Cotton crop is infected by geminiviruses (begomoviruses and mastreviruses) and their associated betasatellites and alphasatellites [3,4]. The analysis of non-cultivated *Gossypium* species suggested that a diverse mixture of geminiviruses and associated satellites infect cotton plants [36]. Here, we presented a detailed picture of the diversity of alphasatellites isolated from cotton and other plants in the last 20 years. CLCMulA was first reported in cotton and ageratum plants; later, several other alphasatellites were discovered [1,8]. The information we collected from the GenBank data suggests that CLCMulA is not only found in cotton plants but also in several other crops, such as tomato, okra, spinach, luffa ipomea and, most surprisingly, sugarcane.

The presence of 12 different alphasatellites in association with cotton leaf curl geminiviruses is an interesting phenomenon despite their non-essential role in disease etiology [9]. In the Old World, the role of alphasatellites has been elucidated as the suppression of gene silencing [15,16], and they attenuate disease symptoms [17,18,19]. Recent experiments conducted on New World alphasatellites [12,14] have suggested that their presence can increase the severity of symptoms. However, the presence of alphasatellites interferes with viral transmission by the whitefly vector. Therefore, it can be speculated that during the initial infection stages, alphasatellites may provide the necessary platform for viral infection by suppressing the natural immunity of plants. However, in the later stages of infection, they might compete with the helper virus and lower its replication level. Their ability to act as suppressors of gene silencing at the transcriptional and post-transcriptional levels may be one reason why diverse alphasatellites are maintained by their helper viruses.

The presence of AEA and CLCMulA in a large number of hosts suggests that there is no natural barrier in different plants that allows for diverse alphasatellite resistance.

The global spread of begomoviruses has been well established [37,38]. The widespread geographical distribution and prevalence of *Colecusatellite* in multiple hosts may cause a problem for world agriculture in the future. International trade may be one reason for their widespread distribution. Future research should focus on the factors that cause the spread of alphasatellites.

Recombination is a major source of genetic diversity among geminiviruses. However, very few alphasatellites have been studied for their inter- or intra-species recombination [26]. To our knowledge, this is the first time the impact of recombination on the genetic diversity of CLCMulA has been analyzed. In our analysis, we identified five different strains of CLCuMuA (strains a, c, e, g and i, each representing the clusters shown in Figure 1) that exhibited recombination with AEA or GDarSLA for different lengths of nucleotide spans. The data based on the recombination analysis suggest that the evolution of CLCMulA, AEA, AYVINA, GDarSLA and CLCLucA occurred due to the exchange of nucleotides with each other. In all the recombinants, the presence of CLCMulA and AEA was evident. From the phylogeny and recombination analysis, it was also clear that CLCLucA evolved locally in India through recombination.

The pairwise sequence comparison revealed that the species criteria set for alphasatellites based on the nucleotide identity percentage at the full-length genome level are valid. However, due to the frequent exchange of nucleotides among CLCMulA, AEA, CLCLucA, AYVINA and GDarSLA, the similarity index of certain isolates is increased. Therefore, we propose that in the future, the molecules that have a high percentage of identity at the alpha-Rep gene level may be merged into a single species.

The mean nucleotide substitution rate of CLCMulA was relatively high (1.56 × 10^−3^), albeit very close to EACMV-CP (1.37 × 10^−3^) and GDarSLA (2.13 × 10^−3^) [36,39]. The nucleotide substitution rate for the different codon positions in the alpha-Rep of CLCMulA was particularly interesting. The nucleotides at the first codon position were highly variable (mean rate of 1.981) compared to the second and third codon positions. Similar codon position substitution rates were observed for CLCMulV-CP and GDarSLA [40]. The higher nucleotide substitution rate at the first codon position is indicative of changing natural selection pressures on alpha-Rep.

## 5. Conclusions

The high genetic diversity of alphasatellites isolated from cotton plants suggests that they do not play a role in disease etiology, and they are still an important component of many disease complexes. Evidently, in the course of the last 20 years, alphasatellites have been identified in a large number of crops and weeds. In certain cases, they act as suppressors of gene silencing. NW alphasatellites have been reported to increase disease symptoms while having a negative impact on viral transmission through the whitefly vector. Taken together, this information suggests that they must play an important biological role in order to be maintained in devastating diseases of crops, including cotton. Furthermore, the factors responsible for their spread across the globe must be controlled to avoid future problems in agriculture.

## Figures and Tables

**Figure 1 pathogens-11-00763-f001:**
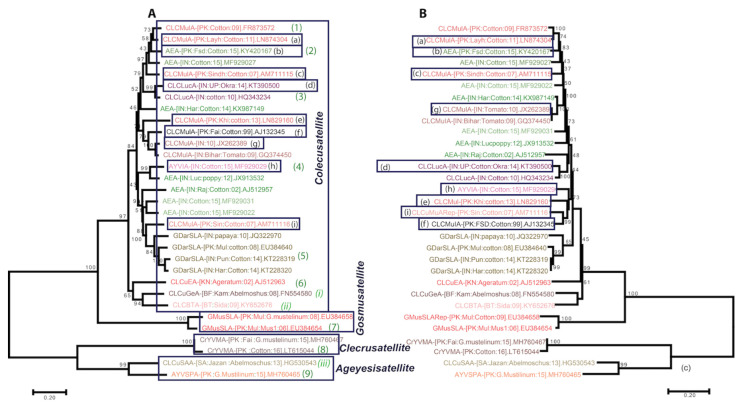
Maximum likelihood phylogenetic tree of alphasatellites isolated from cotton. The numbers at different nodes represent the 1000 bootstrap replication values. Nine different alphasatellites (1)–(9) were reported in cotton crop from around the world, while three alphasatellites (*i*, *ii* and *iii*) were associated only with cotton leaf curl geminiviruses and were not identified in cotton. Panel (**A**) represents the full-length molecules representative of those shown in Appendix A, while panel (**B**) shows the phylogenetic tree of the Rep genes used in panel (**A**). Different clusters representing each strain of CLCMulA are indicated with lowercase letters (a–i) and a rectangular box. Each alphasatellite is represented with a unique color. Members of the genera *Colecusatellite*, *Gosmusatellite*, *Clecrusatellite* and *Ageyesisatellite* infect cotton crops around the world.

**Figure 2 pathogens-11-00763-f002:**
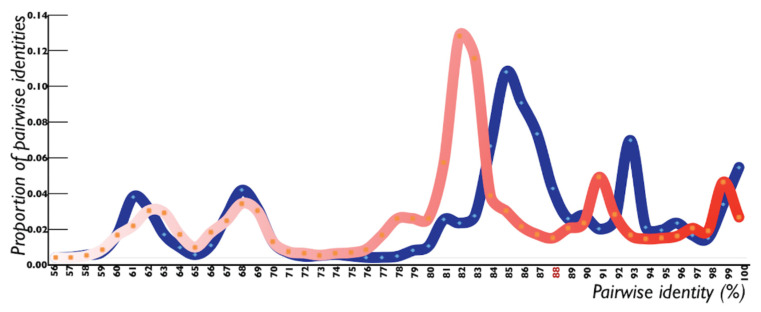
Pairwise sequence comparisons of full-length alphasatellites and their Rep genes. The PASC values of the full-length molecules showed higher levels of nucleotide diversity, even below the species level previously described for alphasatellites (88%). However, the PASC values of the Rep gene were in agreement with the species criteria for alphasatellites. The shift in PASC comparisons was due to recombination in the A-rich or common region of the alphasatellites.

**Table 1 pathogens-11-00763-t001:** Host species mentioned in the GenBank data and classification of geminialphasatellites associated with cotton leaf curl geminiviruses around the world.

Ser#	Alphasatellite	Geographical Distribution	Host Range
Ageyesisatellite			
1	CLCuSAA	Saudi Arabia	Okra
2	AYVSGA	Pakistan	Cotton
Clecrusatellite			
3	CrYVMA	Pakistan India	Cotton, croton, tomato
Colecusatellite			
4	AEA	Pakistan India Oman	Cotton, okra, capsicum, luffa, tomato, sesbinia, hibiscus, sugar cane, jasminum ageratum, hot pepper, populus alba, radish, papaya, poppy, croton, synedrella, qatal booti, mungbean
5	AYVINA	India	Ageratum, cotton, parthenium, amaranthus, radish
6	CLCEA	Egypt Kenya	Cotton, ageratum
7	CLCGezA	Cameroon Saudi Arabia Sudan Burkina Faso Mali Botswana	Okra, tobacco, hollyhock, bamako, sida
8	CLCBTA	Botswana	Sida
9	CLCLucA	India	Okra, cotton
10	CLCMulA	Pakistan India	Cotton, okra, tomato, ipomoea, cyamopsis, saccharum, spinacia, luffa, wheat
11	GDarSLA	India	Cotton, alcea, luffa
		Pakistan	
Gosmusatellite			
12	GMusSLA	India	Cotton, luffa, okra, papaya, alcea
		Pakistan	

Nine different alphasatellites—AYVSGA, CrYVMA, AEA, AYVINA, CLCEA, CLCLucA, CLCMulA, GDarSLA and GMusSLA—were considered bona fide alphasatellites, as they were isolated from cotton. CLCSAA, CLCGezA and CLCBTA have never been found in cotton plants; therefore, we considered them as putative cotton-infecting alphasatellites. Eight out of twelve alphasatellites were present on the Indian subcontinent. The accession numbers of each alphasatellite/host are shown in Appendix A.

**Table 2 pathogens-11-00763-t002:** Nucleotide identity percentages of 12 geminialphasatellites isolated in association with cotton leaf curl geminiviruses.

Alphasat-EllitesName	CLCMulA(Rep)	AEA	CLCLucA	AYVINA	GDarSLA	CLCEA	CrYVMA	AYVSGA	GMusSLA	CLCGezA	CLCBTA	CLCSAA
CLCMulA	84–100 (85–100)											
AEA	76–90 (80–90)	83–100 (89–100)										
CLCLucA	78–88 (81–91)	82–86 (85–89)	93–100 (96–100)									
AYVINA	81–88 (85–91)	80–88 (85–90)	82–83 (86–88)	95–100 (96–100)								
GDarSLA	75–90 (80–89)	76–87 (79–89)	77–81 (80–84)	78–84 (81–88)	88–100 (90–100)							
CLCuEA	73–79 (78–82)	74–88 (79–83)	74–75 (79–80)	77–78 (80–81)	73–77 (77–81)	100 (100)						
CrYVMA	58–63 (58–64)	59–63 (58–63)	59–62 (60–62)	61–62 (60–62)	59–64 (58–64)	60–60 (60–63)	95–100 (90–100)					
AYVSGA	58–62 (58–64)	58–63 (59–64)	58–60 (58–60)	60–62 (60–62)	57–63 (57–64)	59–61 (60–61)	61–63 (61–62)	92–100 (95–100)				
GMusSLA	64–73 (65–72)	63–71 (65–72)	68–72 (66–72)	65–71 (67–70)	62–70 (64–72)	65–69 (66–70)	56–60 (56–62)	58–63 (59–63)	84–100 (87–100)			
CLCGezA	75–81 (79–83)	76–81 (79–83)	76–78 (79–82)	78–80 (81–82)	73–79 (79–83)	75–79 (80–82)	59–63 (60–63)	58–62 (58–62)	62–69 (62–70)	95–100 (97–100)		
CLCBTA	75–79 (79–81)	75–79 (79–82)	77 (79–81)	76–77 (80–81)	72–77 (76–80)	76 (79)	62–63 (62–63)	59–61 (59–61)	64–69 (65–69)	80–82 (83–85)	100 (100)	
CLCSAA	57–63 (56–63)	58–63 (56–63)	59–60 (59–61)	59–62 (60–61)	60–64 (58–64)	58–59 (58)	61–62 (61–62)	77–79 (81–82)	59–63 (59–63)	58–61 (59–64)	59 (60)	100 (100)

The nucleotide identity percentages ranged among the alphasatellites. The values for the full-length genomes of alphasatellites showed a wider range of identity percentages due to recombination, deletions or insertions, while the identity percentage values for the alpha-Rep genes (in parentheses) showed a higher level of similarity.

**Table 3 pathogens-11-00763-t003:** Nucleotide substitution rate estimates for cotton leaf curl Multan alphasatellites.

Satellite/Virus Acronym	Demographic/Clock Model	Mean Rate	CP1 Mutation Rate	CP2 Mutation Rate	CP3 Mutation Rate	No. of Taxa
GDarSLA-Rep *	BSP/Relaxed	2.13 × 10^−3^	1.4	0.765	0.831	63
CLCMulB-βC1 *	BSP/Relaxed	3.51 × 10^−3^	0.85	0.73	1.43	39
CLCMulA-Rep	GTR/Relaxed	1.56 × 10^−3^	1.981	0.516	0.501	36
ChiLCA-Rep **	-	2.25 × 10^−3^	-	-	-	-

* Nawaz-ul-Rehman et al., 2012; ** Kumar et al., 2017.

## Data Availability

Data supporting reported results could be found with corresponding author.

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
