# Peer review of "Patterns of Genetic Diversity among Alphasatellites Infecting Gossypium Species"

_pathogens, 2022, doi:10.3390/pathogens11070763_

Round 1

Reviewer 1 Report

This is a resubmission of a manuscript that I have previously reviewed, and recommended rejection due to a number of experimental flaws and mis/overinterpretation of the results. This new version is considerably improved, but still needs a considerable amount of work.

General (major) comments:

a) In the Introduction, the distinction between OW and NW alphasatellites is mentioned (L48-49) but not explained. They are not only phylogenetically distinct (in fact, they are classified in different genera), but there are also differences in terms of their role in disease, with OW alphas not affecting symptoms, while NW alphas are associated with an increase in symptom severity (Paprotka et al., 2010, Mar et al. 2017, Nogueira et al. 2021). This should be mentioned here.

b) The next paragraph of the Introduction (L52-60) is very poorly written. First, all alphas (from the three subfamilies) are autonomously replicating molecules. alphaRep has indeed been reported as a suppressor of PTGS, but has also been reported as a suppressor of *TGS* (Abbas et al., VirusDisease 2017, doi 10.1007/s13337-017-0413-5). The paper by Mar et al. does indeed report a negative effect of the alphasat on vector transmission, but it is a preliminary result based on a small number of plants. Nogueira et al. obtained the same result for a different begomovirus/alphasat combination and with a larger number of plants. As mentioned above, OW alphas do not generally affect symptoms. However, there is one report of an "unusual" alphasatellite now classified in the genus Ageyesysatellite) and one other report from China of alphas attenuating symptoms, and the above-mentioned reports of NW alphas increasing symptoms.  

c) The authors still struggle to explain the rationale for the study. In L71-73, the references cited do not really provide support for the authors' claim. In the first reference, several alphas were found in cotton (contrary to what is stated), and Ref. 26 is a review. As I mentioned in my previous review, the supposed increase in the number of alphas reported from cotton is likely due to the increase in the number of research groups interested in these agents (a classic case of "seek and ye shall find"). Furthermore, the etiology of CLCD has been established a long time ag and alphas play no role, so the statement in L76-77 is incorrect (in fact, it is directly contradicted by the following sentence). Again, as mentioned in my previous review, a manuscript that advances our understanding of alphasatellites and their interaction with their helper viruses would be welcome. This new version of the manuscript is definitely a step in the right direction and the authors state quite clearly its objectives: to perform a diversity and genetic variability analysis of alphasatellites based on sequences deposited in GenBank (L86-88). They need to drop the "increase in diversity" story once and for all.

d) The recombination analysis is better explained but one concern is the statement in L117-118. Did the authors use only the GENECONV module ? This is not good practice. The reliability of the recombination events detected by RDP is better assessed by comparing the results of all modules. Reliable events should be detected by all of them, and in most studies the authors only report events detected by at least four modules. In my previous review I criticized the figure (which was copy/pasted from RDP itself). The figure in this manuscript is not much different ! It is basically a redraw of the RDP figure. My suggestion is that the authors delete the figure and present the recombination results in a table, listing each event in a separate line and including the major and minor parents, the recombination breakpoints, the number of RDP modules that detected the event and the lowest p-value obtained. This is much more informative. If they absolutely want to keep the figure, move it to supplementary material.  

e) The authors continue to misinterpret the phylogenetic analysis. It does *not* show that alphasatellites infect cotton ! A phylogenetic tree informs ancestry and evolutionary relationships. That this or that alphasat infects cotton is informed from the GenBank metadata. The Results section should start descriptively. Delete the first two paragraphs and start with the text from L163 and with Table 1. The phylogenetic tree should be moved to section 3.2. In this section, delete the first sentence (a phylogenetic tree does not reveal diversity !). On the other hand, I agree with the authors that the trees suggest recombination between AEA and CLCMulA (but phylogenetic analysis alone does not prove it; this comes with the RDP result). In fact, the authors' analysis does indicate that the taxonomic status of these two species should be reevaluated, and *maybe* they should be merged. The authors could do this by applying the "conflict resolution criteria" established by Briddon et al. 2018 (see page 5129 of that paper).  

f) Section 3.3 is problematic. Its first paragraph again misinterprets the phylogenetic analysis and should be deleted. The second paragraph seems to be an attempt to discuss the taxonomic status of some alphasatellites, but if this is the case, it is poorly done. Figure 2 does not show percent identity data for individual species, so referring to this figure in L223 is incorrect. The fact CLCMulA, AEA and GMusSLA isolates may have identities below 88% is no problem at all according to the above-mentioned conflict resolution criteria. On the other hand, the fact that isolates of CLCMulA and AEA have identities ranging from 76 to 90% suggests that these two species should be merged. The same is true for CLCMulA and GDarSLA - their isolates have identities ranging from 75 to 90%. The focus of section 3.3 on the whole genome vs Rep identities is misguided. Species demarcation based on whole genome nucleotide sequences is solid, and unlikely to be changed by the ICTV. But the taxonomy status of CLCMulA, AEA and GDarSLA as distinct species does seem to be incorrect, ad this section would be the place to discuss it and propose that the three species should be merged.

g) The Discussion needs to be rewritten along the major points raised above. This is a genetic variability analysis paper, and the discussion should focus on that. So, for example, the text in L344-349 should be deleted.

h) The language of the manuscript continues to be a problem, with numerous spelling mistakes, grammatical and syntax errors, incorrect use of articles, capitalization, etc. A revised version must be proofread by a native English speaker or by a commercial proofreading service (I have no affiliation whatsoever with any commercial service and the reason why I am mentioning them is because I have used them myself and found them to be quite efficient).

Specific comments:

The correct way to write the name of the protein is alpha-Rep (not Alpha-Rep).

L51: Reference 10 reports the association of two alphasatellites with a nanovirus, *not* with a bipartite begomovirus.  

L69: There are 85 alphasatellite species (not ~85).

L83-84: This sentence does not make sense.

L156-157: This is misleading. The way the sentence is written ("previously only limited to weeds") suggests that the host range of these alphas has somehow increased to include cotton. As in the previous version of the manuscript, this is a misinterpretation of the data. The agents were simply detected in weeds first, and then, later, were detected in cotton. The sentence should be deleted.

L153-154: The correct name is cotton leaf curl Egypt alphasatellite. Here the authors are referring to alphasatellites, not to species (so no italics). A species does not infect anything, since it is an abstract category.

L163-164: Suggestion: The subfamily Geminialphasatellitinae comprises of seven genera. Members of four of these genera (Ageyesisatellite, Clecrusatellite, Colecusatellite and Gosmusatellite) infect cotton in Asia and Africa (Table-1).

In Table 1, remove the bullets before the country names, and the uppercase letters in the host names. I strongly suggest aligning the columns to the left instead of in the center.

In Table 2, reduce the size of the font so that all acronyms fit in a single line.

L211-214: This is overinterpretation of the data and should be deleted.

Author Response

Comments and Suggestions for Author 1

This is a resubmission of a manuscript that I have previously reviewed, and recommended rejection due to a number of experimental flaws and mis/overinterpretation of the results. This new version is considerably improved, but still needs a considerable amount of work.

Ans: We are thankful for the reviewer, as the manuscript was read in full depth and the suggestions were very positive.

General (major) comments:

  1. a) In the Introduction, the distinction between OW and NW alphasatellites is mentioned (L48-49) but not explained. They are not only phylogenetically distinct (in fact, they are classified in different genera), but there are also differences in terms of their role in disease, with OW alphas not affecting symptoms, while NW alphas are associated with an increase in symptom severity (Paprotka et al., 2010, Mar et al. 2017, Nogueira et al. 2021). This should be mentioned here.

Ans: We agreed with the reviewer and the suggested information was added.

  1. b) The next paragraph of the Introduction (L52-60) is very poorly written. First, all alphas (from the three subfamilies) are autonomously replicating molecules. alphaRep has indeed been reported as a suppressor of PTGS, but has also been reported as a suppressor of *TGS* (Abbas et al., VirusDisease 2017, doi 10.1007/s13337-017-0413-5). The paper by Mar et al. does indeed report a negative effect of the alphasat on vector transmission, but it is a preliminary result based on a small number of plants. Nogueira et al. obtained the same result for a different begomovirus/alphasat combination and with a larger number of plants. As mentioned above, OW alphas do not generally affect symptoms. However, there is one report of an "unusual" alphasatellite now classified in the genus Ageyesysatellite) and one other report from China of alphas attenuating symptoms, and the above-mentioned reports of NW alphas increasing symptoms.  

Ans: We modified the text accordingly. The role of Old World and New World alphasatellites was added alongwith the references mentioned by the reviewer.

  1. c) The authors still struggle to explain the rationale for the study. In L71-73, the references cited do not really provide support for the authors' claim. In the first reference, several alphas were found in cotton (contrary to what is stated), and Ref. 26 is a review. As I mentioned in my previous review, the supposed increase in the number of alphas reported from cotton is likely due to the increase in the number of research groups interested in these agents (a classic case of "seek and ye shall find"). Furthermore, the etiology of CLCD has been established a long time ago and alphas play no role, so the statement in L76-77 is incorrect (in fact, it is directly contradicted by the following sentence). Again, as mentioned in my previous review, a manuscript that advances our understanding of alphasatellites and their interaction with their helper viruses would be welcome. This new version of the manuscript is definitely a step in the right direction and the authors state quite clearly its objectives: to perform a diversity and genetic variability analysis of alphasatellites based on sequences deposited in GenBank (L86-88). They need to drop the "increase in diversity" story once and for all.

Ans: Indeed, this manuscript is about the genetic diversity of alphasatellites isolated from cotton over a period of two decades. Once you get an infected sample and you clone several satellites/viruses out of it. With the increase in cloning experiments one can expect more number of sequences. But this is not always true that whenever you will analyze a sample a different molecule will be there.

  1. d) The recombination analysis is better explained but one concern is the statement in L117-118. Did the authors use only the GENECONV module ? This is not good practice. The reliability of the recombination events detected by RDP is better assessed by comparing the results of all modules. Reliable events should be detected by all of them, and in most studies the authors only report events detected by at least four modules. In my previous review I criticized the figure (which was copy/pasted from RDP itself). The figure in this manuscript is not much different ! It is basically a redraw of the RDP figure. My suggestion is that the authors delete the figure and present the recombination results in a table, listing each event in a separate line and including the major and minor parents, the recombination breakpoints, the number of RDP modules that detected the event and the lowest p-value obtained. This is much more informative. If they absolutely want to keep the figure, move it to supplementary material.  

Ans: We used 6 different modules to detect the recombinants and we mentioned it in the material and methods section now. Previously, the same criticized appeared that direct outcome of RDP should not be there. This time, we modified the diagram with more labelling. In my view the outcome of RDP should be directly presented. However, this time we moved the figure-3 to supplementary files. The breakpoints and p values are provided in the supplementary.

  1. e) The authors continue to misinterpret the phylogenetic analysis. It does *not* show that alphasatellites infect cotton ! A phylogenetic tree informs ancestry and evolutionary relationships. That this or that alphasat infects cotton is informed from the GenBank metadata. The Results section should start descriptively. Delete the first two paragraphs and start with the text from L163 and with Table 1. The phylogenetic tree should be moved to section 3.2. In this section, delete the first sentence (a phylogenetic tree does not reveal diversity !). On the other hand, I agree with the authors that the trees suggest recombination between AEA and CLCMulA (but phylogenetic analysis alone does not prove it; this comes with the RDP result). In fact, the authors' analysis does indicate that the taxonomic status of these two species should be reevaluated, and *maybe* they should be merged. The authors could do this by applying the "conflict resolution criteria" established by Briddon et al. 2018 (see page 5129 of that paper).  

Ans: We re-wrote the results and modified as suggested. The phylogenetic tree is moved to section 3.2. At this moment, we suggested that AEA or CLCMulA “may be” merged. As mentioned by the reviewer that full length alplhasatellites are the best criteria for taxonomy, it is clear that Rep alone can not be used for classification of alphasatellites.

  1. f) Section 3.3 is problematic. Its first paragraph again misinterprets the phylogenetic analysis and should be deleted. The second paragraph seems to be an attempt to discuss the taxonomic status of some alphasatellites, but if this is the case, it is poorly done. Figure 2 does not show percent identity data for individual species, so referring to this figure in L223 is incorrect. The fact CLCMulA, AEA and GMusSLA isolates may have identities below 88% is no problem at all according to the above-mentioned conflict resolution criteria. On the other hand, the fact that isolates of CLCMulA and AEA have identities ranging from 76 to 90% suggests that these two species should be merged. The same is true for CLCMulA and GDarSLA - their isolates have identities ranging from 75 to 90%. The focus of section 3.3 on the whole genome vs Rep identities is misguided. Species demarcation based on whole genome nucleotide sequences is solid, and unlikely to be changed by the ICTV. But the taxonomy status of CLCMulA, AEA and GDarSLA as distinct species does seem to be incorrect, ad this section would be the place to discuss it and propose that the three species should be merged.

 Ans: The interpretation of phylogenetic tree from section 3.3 is removed. The figure-2 represents the PASC values. We discussed here that rep genes of some molecules like, CLCMulA, AEA and GDarSLA are remarkably similar, therefore their taxonomic status may be revised in the future.

  1. g) The Discussion needs to be rewritten along the major points raised above. This is a genetic variability analysis paper, and the discussion should focus on that. So, for example, the text in L344-349 should be deleted.

Ans: We rephrased several points raised by the reviewer and it is modified now.

  1. h) The language of the manuscript continues to be a problem, with numerous spelling mistakes, grammatical and syntax errors, incorrect use of articles, capitalization, etc. A revised version must be proofread by a native English speaker or by a commercial proofreading service (I have no affiliation whatsoever with any commercial service and the reason why I am mentioning them is because I have used them myself and found them to be quite efficient).

 Ans: The language of the manuscript has been corrected by MDPI English language experts. We hope that it will not be problematic anymore here.

Specific comments:

The correct way to write the name of the protein is alpha-Rep (not Alpha-Rep).

Ans. Corrected

L51: Reference 10 reports the association of two alphasatellites with a nanovirus, *not* with a bipartite begomovirus.  

Ans: The proper reference is added.

L69: There are 85 alphasatellite species (not ~85).

L83-84: This sentence does not make sense.

Ans; Both are corrected.

L156-157: This is misleading. The way the sentence is written ("previously only limited to weeds") suggests that the host range of these alphas has somehow increased to include cotton. As in the previous version of the manuscript, this is a misinterpretation of the data. The agents were simply detected in weeds first, and then, later, were detected in cotton. The sentence should be deleted.

Ans: We deleted the sentenced.

L153-154: The correct name is cotton leaf curl Egypt alphasatellite. Here the authors are referring to alphasatellites, not to species (so no italics). A species does not infect anything, since it is an abstract category.

Ans. Corrected.

L163-164: Suggestion: The subfamily Geminialphasatellitinae comprises of seven genera. Members of four of these genera (AgeyesisatelliteClecrusatelliteColecusatellite and Gosmusatellite) infect cotton in Asia and Africa (Table-1).

Ans; Corrected, as suggested.

In Table 1, remove the bullets before the country names, and the uppercase letters in the host names. I strongly suggest aligning the columns to the left instead of in the center.

In Table 2, reduce the size of the font so that all acronyms fit in a single line.

Ans: Both the tables are modified now.

L211-214: This is overinterpretation of the data and should be deleted.

Ans: We deleted the sentences.

Reviewer 2 Report

In the submitted manuscript, Mubin et al. analyze the sequence diversity of Alphasatellites infecting Gossypium spp. based on a meta-analysis of sequences. The authors present the sequence diversity, analyze the evolutionary relationship, and identify crossing over events. While this paper is interesting, and many of the methods appear to be appropriate, there are a number of detracting factors that make the analysis and conclusions impossible to follow. Figures are incomplete. Graph axis are not labelled (fig 2). phylogenies are overly small, labelled with multiple different signifiers like colour, boxes and other minutiae that are not explained in the legend (Fig 1). The table contains random bullet points for no reason. The introduction, while generally being reasonable, does have a few occurrences of crude writing that are difficult to follow (ln48-50). The results section can be abrupt and hard to follow.  Many of the results are interesting, including the nucleotide sequence diversity analysis, and recombination analysis, and there is a large amount of interesting information in this manuscript relating to alpha satellites. However, presentation of the Figure 1 and 2 need to be dramatically improved before full comprehension is possible. 

Minor points:

-revised text has errors in it ln 42 "realtively"

Author Response

Comments and Suggestions for Authors

In the submitted manuscript, Mubin et al. analyze the sequence diversity of Alphasatellites infecting Gossypium spp. based on a meta-analysis of sequences. The authors present the sequence diversity, analyze the evolutionary relationship, and identify crossing over events. While this paper is interesting, and many of the methods appear to be appropriate, there are a number of detracting factors that make the analysis and conclusions impossible to follow. Figures are incomplete. Graph axis are not labelled (fig 2). phylogenies are overly small, labelled with multiple different signifiers like colour, boxes and other minutiae that are not explained in the legend (Fig 1). The table contains random bullet points for no reason. The introduction, while generally being reasonable, does have a few occurrences of crude writing that are difficult to follow (ln48-50). The results section can be abrupt and hard to follow.  Many of the results are interesting, including the nucleotide sequence diversity analysis, and recombination analysis, and there is a large amount of interesting information in this manuscript relating to alpha satellites. However, presentation of the Figure 1 and 2 need to be dramatically improved before full comprehension is possible. 

 Ans: We are thankful for the reviewer for raising important concerns. The figure legends for figure-1 are modified as suggested. While, figure-2 is also modified with graph labelling. We hope that it will not be a problem.

Regarding the results section, it is modified extensively as suggested by the reviewer-1.

Minor points:

-revised text has errors in it ln 42 "realtively"

Ans: Corrected.

Round 2

Reviewer 1 Report

I commend the authors for following the comments and suggestions made on the previous version of this manuscript - this revised version is vastly superior. All the major issues were properly addressed and the manuscript definitely reads much better, with a logical flow of information and no over-interpretation of results. At this point, only one major issue remains that will require the authors' attention: the incorrect reference to "species". As explained by the ICTV in several papers (the most recent being Zerbini et al., Arch Virol 167:1231, 2022), a species is an abstract concept and therefore does not have a genome, cannot be isolated and does not cause disease. These are all properties of viruses and, of course, alphasatellites. So there are several instances in the text where the reference to species must be dropped (I highlighted most of them in the edited pdf, but I may have missed some). In all of these instances the authors should replace "species" with "alphasatellites". Other than that, I provide only editorial corrections (please see the attached pdf). This manuscript has come a long way, and at this point it requires mostly minor revisions before it can be accepted for publication. 

Author Response

I commend the authors for following the comments and suggestions made on the previous version of this manuscript - this revised version is vastly superior. All the major issues were properly addressed and the manuscript definitely reads much better, with a logical flow of information and no over-interpretation of results.

We are thankful to the reviewer for nice comments.

At this point, only one major issue remains that will require the authors' attention: the incorrect reference to "species". As explained by the ICTV in several papers (the most recent being Zerbini et al., Arch Virol 167:1231, 2022), a species is an abstract concept and therefore does not have a genome, cannot be isolated and does not cause disease. These are all properties of viruses and, of course, alphasatellites. So there are several instances in the text where the reference to species must be dropped (I highlighted most of them in the edited pdf, but I may have missed some). In all of these instances the authors should replace "species" with "alphasatellites". Other than that, I provide only editorial corrections (please see the attached pdf). This manuscript has come a long way, and at this point it requires mostly minor revisions before it can be accepted for publication.

Indeed, we agreed with the reviewer and replaced the word species by “alphasatellites” as suggested by the reviewer.

Reviewer 2 Report

The authors have addressed issues raised previously.

Author Response

The reviewer-2 has Kindly accepted our manuscript in its current format. However, the modified version while keeping in view the previous comments is provided.